# Effect of Covering a Visitor Viewing Area Window on the Behaviour of Zoo-Housed Little Penguins (*Eudyptula minor*)

**DOI:** 10.3390/ani10071224

**Published:** 2020-07-18

**Authors:** Samantha J. Chiew, Kym L. Butler, Sally L. Sherwen, Grahame J. Coleman, Vicky Melfi, Alicia Burns, Paul H. Hemsworth

**Affiliations:** 1Animal Welfare Science Centre, Faculty of Veterinary and Agricultural Sciences, University of Melbourne, Parkville, VIC 3052, Australia; kym.butler@unimelb.edu.au (K.L.B.); grahame.coleman@unimelb.edu.au (G.J.C.); phh@unimelb.edu.au (P.H.H.); 2Biometrics Team, Agriculture Victoria Research, Department of Jobs, Precincts and Regions, Hamilton, VIC 3300, Australia; 3Department of Wildlife Conservation and Science, Zoos Victoria, Parkville, VIC 3052, Australia; ssherwen@zoo.org.au; 4Department of Animal and Agriculture, Hartpury University and Hartpury College, Gloucester GL19 3BE, UK; Victoria.melfi@hartpury.ac.uk; 5Taronga Institute of Science and Learning, Taronga Conservation Society Australia, Mosman, NSW 2088, Australia; aburns@zoo.nsw.gov.au; 6School of Life and Environmental Sciences, University of Sydney, Sydney, NSW 2006, Australia

**Keywords:** little penguins, visitor-animal interactions, visual contact, zoos, visitor effect, visitor behaviour

## Abstract

**Simple Summary:**

Penguins are a common zoo-housed species and have been shown to display behaviours indicative of fear such as huddling, vigilance and avoidance towards zoo visitors. However, this evidence has been obtained from a single public zoo in Melbourne, Australia. Therefore, we investigated the effect of covering a visitor viewing area window on fear behaviour of zoo-housed little penguins at another zoo in Sydney, Australia. Covering one out of four visitor viewing area windows reduced the number of visitors and the occurrence of potentially threatening visitor behaviours at this window such as banging on the window, loud vocalisations and sudden movement. When the viewing window was covered, the number of penguins visible and preening in the water increased and the number of penguins vigilant near this viewing window reduced. Also, the adjacent corner area, which was not visible to visitors, was found to be a preferred area for the penguins whether the viewing window was uncovered or covered. While there were limited effects, the reduced presence, reduced preening in the water and increased vigilance by penguins near the viewing window when this window was uncovered, together with the general preference for the corner area, provides evidence of some avoidance of visitors. These results suggest that visual contact with visitors and/or other types of visitor contact, such as visitor-induced sounds and vibrations, may be fear-provoking for zoo-housed little penguins. Therefore, these results suggest that penguins in zoos may benefit from modifications to the enclosure that may ameliorate penguin fear responses to visitors such as one-way viewing glass, barriers reducing close visitor contact and areas for penguins to retreat.

**Abstract:**

Studies on the effects of visitors on zoo animals have shown mixed findings and as a result, the manner in which visitors affect zoo animals remains unclear for many species, including a rarely studied taxa such as penguins. Penguins are a common zoo-housed species and have been shown to display huddling, vigilance and avoidance towards zoo visitors which can be indicative of fear. Here, we examined the effects of covering one visitor viewing area window, out of four, on little penguin (*Eudyptula minor*) behaviours that may be indicative of fear. Two treatments were randomly imposed on different days: (1) The main visitor viewing area window, where most visitor-penguin interactions occurred, was uncovered (‘Main window uncovered’) and (2) The main visitor viewing area window was covered (‘Main window covered’). Penguin numbers and behaviour were recorded near the main visitor viewing area window and the three other visitor viewing area windows, as well as one area not visible to visitors (‘Corner’ area). Furthermore, visitor numbers and visitor behaviour were recorded at all four visitor viewing area windows. Covering the main visitor viewing area window reduced the proportion of visitors present at this window by about 85% (*p* < 0.001) and reduced potentially threatening visitor behaviours at this window such as tactile contact with the window, loud vocalisations and sudden movement (*p* < 0.05). When the main visitor viewing area window was covered, the proportion of penguins present increased by about 25% (*p* < 0.05), the proportion of visible penguins preening in the water increased by about 180% (*p* < 0.05) and the proportion of visible penguins vigilant decreased by about 70% (*p* < 0.05) in the area near this main window. A preference for the Corner area was also found whereby 59% and 49% of penguins were present in this area when the main window was uncovered and covered, respectively. These results provide limited evidence that the little penguins in this exhibit showed an aversion to the area near the main visitor viewing area window when it was uncovered based on the increased avoidance and vigilance and decreased preening in the water in this area. This suggests visitors may be fear-provoking for these little penguins. However, it is unclear whether visual contact with visitors *per se* or other aspects of visitor contact, such as visitor-induced sounds and vibrations, were responsible for this apparent aversion when this window was uncovered.

## 1. Introduction

Research on the effect of visitors on zoo animals has expanded significantly in the past several years, providing increasing evidence of the various effects visitors can have on a range of zoo species [1]. However, the literature presents mixed findings, possibly because of a number of factors that may affect the response of zoo animals to visitors such as species-specific differences, individual animal characteristics, enclosure design and the nature and intensity of visitor-animal interactions [1,2]. As a result, the manner in which visitors affect zoo animals still remains unclear for many species, especially for a rarely studied taxa such as penguins [3]. Understanding the way visitors affect zoo animals is important because visitor contact can be unpredictable and intense, with zoo animals being exposed to a range of stimuli from visitors including auditory, visual, tactile, olfactory and vibratory stimuli [1,4]. These visitor stimuli may be perceived by zoo animals as, for example threatening or stimulating, and so understanding this can help identify characteristics of visitors, animals and/or enclosures that affect the nature of visitor contact and be subsequently managed accordingly by zoos [1,3]. Furthermore, it is important for zoos to also understand how zoo animals affect visitors as there is growing evidence to show that zoo animals, particularly their behaviour, can affect visitor attitudes, behaviour and experience as well as zoo conservation efforts [5,6,7,8,9]. Consequently, understanding zoo visitor-animal interactions has never been more pertinent.

Limited research has been conducted to examine the effects of visitors on zoo-housed penguins despite penguins being a commonly housed zoo species, especially little penguins (*Eudyptula minor*) in Australian zoos [3,10,11,12,13,14,15]. From the limited number of studies that have investigated the zoo visitor-penguin relationship, only two studies at Melbourne Zoo (Australia) have been conducted under experimental conditions [3,10]. The presence of visitors has been found to be fear-provoking for little penguins as indicated by an increased vigilance, huddling and avoidance of the visitor viewing area and reductions in swimming [10]. More recently, Chiew and colleagues [3] identified that it is the close viewing proximity of visitors that increased vigilance, huddling and avoidance of the visitor viewing area and decreased surface swimming in little penguins. These two experiments indicate visual contact with visitors may be fear-provoking for little penguins in zoos but, if the aversion and thus perception of visitors as threatening to little penguins is reduced, this can have beneficial effects on penguin welfare. Possible practical ways this can be achieved is through the use of one-way visual barriers or barriers that prevent visitors getting close to the enclosure which minimises the perceived close proximity of visitors and thus, fear of visitors in penguins [3].

Good vision is important for all penguin species as they are visual predators that rely on their vision to find and catch prey underwater and avoid predators [16,17,18,19,20,21]. Therefore, visual contact maybe a key component in penguin responses to zoo visitors. This can be tested by manipulating the animal’s visual contact with visitors. Only a handful of studies on zoo-housed nonhuman primates have investigated the effects of manipulating visual contact between zoo animals and visitors. For example, reducing visual contact with visitors by using camouflage netting or one-way vision screens has been found to reduced intragroup aggression and stereotypic behaviours such as repetitive teeth clenching, body rocking, pacing and self-scratching in Western lowland gorilla (*Gorilla gorilla gorilla*) and Black-capped capuchins (*Sapajus apella*) [22,23,24]. In contrast, Bloomfield, et al. [25] found that when half of the visitor viewing window was covered with a visual barrier and the other half was left open, orangutans (*Pongo pygmaeus abelii*) showed a preference to position themselves in front of the open visitor viewing window which suggests orangutans are visually stimulated by visitors. These studies indicate the effects of visual contact with visitors can vary between species of nonhuman primates but also show that other forms of visitor contact may have been altered by the visual barriers such as visitor behaviour. While this has yet to be examined in zoo-housed penguins, these studies have highlighted some practical ways visitor-animal interactions can be regulated to identify and understand the way visitors affect penguin behavioural responses in zoos.

There is also some evidence in the wild penguin literature that suggests that penguins of the same species at different sites can be affected by humans in a similar manner. For example, Adélie penguins (*Pygoscelis adeliae*) [26,27,28,29,30]; African penguins (*Spheniscus demersus*) [31,32,33], and little penguins [34,35,36,37] have consistently shown, at various breeding sites, that human disturbances such as human approach and unregulated tourism (i.e., visitor numbers, noise, behaviour and interactions with penguins were uncontrolled) increase heart rates and avoidance of nesting near visitor footpaths and reduce breeding success. In contrast, wild Yellow-eyed penguins (*Megadyptes antipodes*) at different colony sites vary in their response to human disturbances. For example, at some colony sites unregulated tourism reduced reproductive rates, increased corticosterone concentrations and delayed landing times [38,39,40], while no effects on breeding success, heart rate and avoidance behaviour were found at other colony sties [41,42]. These studies on both wild and zoo penguins highlight the importance of understanding the effect of visitors on the same species of penguins at different zoo sites as there may be similarities or variation in their responses which may be a result of the differing nature of visitor contact that occurs because of different environment characteristics. 

The present experiment at Taronga Zoo (New South Wales, Australia) examined the effects of covering one out of four visitor viewing area windows, using a visual barrier, on little penguin behaviours indicative of fear such as avoidance, huddling and vigilance. It was hypothesised that covering the main visitor viewing area window, where most visitor-penguin interactions occurred, would minimise little penguin fear responses toward visitors. It is recognized that other forms of visitor contact may be affected by the visual barrier such as visitor-induced sounds and vibrations.

## 2. Materials and Methods

### 2.1. Study Spcies, Housing and Husbandry

This experiment was approved by the Taronga Conservation Society Australia’s Animal Ethics Committee (approval number 3b/02/17) and was conducted at the Taronga Zoo’s Great Southern Oceans penguin exhibit during March 2017. Little penguins were housed in an outdoor, naturalistic exhibit that consisted of two sections: the main section where 39 adult little penguins were located (breeding group) and the encounter section which contained 13 individuals (nine females and four males; a mix of juveniles and adults aged between 1 to 5 years old) that were involved in daily close-up encounters with visitors (Figure 1). These two sections were adjacent to each other and were separated by a meshed water gate that allowed the penguins in the two groups to have visual contact with one another when in the pool but, had no access to each other.

The encounter penguins were the focus of our experiment as they were identified, through preliminary observations, to be the penguins that visitors stopped to view and interact with the most (Figure 2). Between the 10–15 March 2017, there were a total of 14 little penguins in the encounter section of the exhibit as a new male penguin was introduced into the group. However, this penguin was not eating while in this section and was removed on the 16th March 2017. The encounter penguins were fed twice a day, at approximately 09:00 and 14:00 h, where the 14:00 h feed time was when a close-up encounter with visitors occurred. This involved a maximum of four visitors in the exhibit feeding the penguins with a keeper present. The duration of each encounter was approximately 20–25 min and occurred consistently throughout the study period.

The encounter section of the exhibit was 28 m in length and 2.7 m wide at the narrowest end (where keepers entered/exited the exhibit) to 6.5 m wide at the widest end. It consisted of a 26 m length pool that was long and narrow (1 m wide at narrowest point, 4.3 m at widest point) and went from shallow (0.2 m) to deep water (1.6 m; Figure 2). The visitor pathway (21 m in length, 2.3 m wide) ran along one side of the exhibit, with viewing positions being along the length of the pool (Figure 2). The visitor viewing area of the encounter section of the exhibit consisted of four two-pane glass viewing windows (1.8 m in height) where visitors were able to view the penguins, with the last two glass viewing windows (i.e., Viewing window 3 and 4) having visibility of land areas consisting of sand and vegetation (Figure 2). 

Husbandry, including monitoring, cleaning and feeding of penguins followed normal routines and remained consistent throughout the course of the experiment. Both the breeding group of adult penguins and the encounter penguins were not breeding during the study period, and moulting was completed prior to the study period.

### 2.2. Design & Treatments

We used a fully randomised design, in which two treatments were imposed:(1)Main window uncovered—the visitor viewing area window where most visitor-penguin interactions occurred (the ‘Main visitor viewing area window’), determined by preliminary observations, was unaltered so that a normal view between visitors and penguins was provided. This window was adjacent to the area in which penguins spent most of their time, the deep corner area of the pool (i.e., ‘Corner’; Figure 1).(2)Main window covered—a visual barrier, made up of eight corflute (corrugated polypropylene) panels, was placed on the main visitor viewing area window and eliminated all contact between penguins and visitors at this window (Figure 2 and Figure 3).

Each treatment was randomly imposed for 1-day periods, with 5-replicates of each treatment (total of 10 study days) over 2 weeks. One-day periods were selected because Sherwen et al. [10] found significant changes in penguin behaviour when treatments were imposed for 1-day periods. To allow the penguins to acclimatise to the visual barrier, it was placed on the main visitor viewing area window the afternoon before the scheduled treatment day. The experiment was conducted from the 6 to 20 March 2017 (Autumn) on school-working days, to avoid the normal systematic variation in visitor numbers that occurs on weekends and school holidays.

### 2.3. Animal Behavioural Observations

Five GoPro cameras (Hero 3, GoPro, Inc. San Mateo, CA, USA) were placed around the exhibit which covered five main areas of the enclosure utilised by the penguins and continuously recorded penguin behaviour between 09:30–15:15 h. These areas were adjacent to each other and were labelled: ‘Corner’, ‘Main visitor viewing area window’; ‘Viewing window 2’; ‘Viewing window 3’; and ‘Viewing window 4’ (Figure 4 and Figure 5). These areas included both parts of land and pool except for the Corner area which consisted of only the pool with no land areas. Keepers entered and exited at the Viewing window 4 end of the exhibit which was where the penguins were also fed on land (Figure 5). Also, the Corner area which was on the opposite end to Viewing window 4, was located at the deep end of the pool beyond the main visitor viewing area window, and was not visible to visitors (all other areas were visible to visitors) whether the visual barrier was in place or not (Figure 1). 

The main visitor viewing area window was the window that was manipulated in this experiment using a visual barrier (Figure 1, Figure 2 and Figure 3). Cameras were positioned to capture the field of view from the visitor viewing area windows (Figure 4 and Figure 5). Footage from these areas were used to transcribed penguin behaviour using the VLC Media Player 2.2.1. Observations were only conducted on penguins visible in the field of view of the cameras and individual penguin identity was not recorded because individuals were not easily identifiable. 

Observations were conducted on all study days in 3 × 1 h observation blocks and 1 × 30 min block (10:00–11:00, 11:00–12:00, 12:30–13:30, 14:45–15:15 h), using a combination of instantaneous point sampling for behavioural states and one-zero sampling for behavioural events. It should be noted that the closeup feeding encounter occurred consistently throughout the study period and the fourth observation block was conducted at least 20 min after the encounter to avoid any immediate effects of the encounter on penguin behaviour. Instantaneous point sampling at 3 min intervals was used to record the behavioural states of penguins present in each area and one-zero sampling for 30 s periods, within each 3 min interval, was used to record the number of penguins performing each of the behavioural events (described in Table 1). 

These sampling techniques were chosen as they have been effectively used in a previous study [3,43]. It should be noted that the Corner area was the only location in the exhibit where there was no land area on the other side of the pool for the penguins (Figure 1 and Figure 4). All behaviours were mutually exclusive from one another with the exception of ‘penguins visible’ and ‘huddling (land)’ (Table 1).

### 2.4. Visitor Behavioural Observations

Visitor observations were conducted directly by the principal investigator (SJC) between 09:30–15:15 h in six 30 min observation blocks. Ambient noise level (dB) in the visitor viewing area was logged continuously, using the Mint Muse Sound Meter Pro iPhone app (Mint Muse, Inc., San Diego, CA, USA) at the enclosure in the same observation blocks from the visitor viewing area. Instantaneous point sampling at 3 min intervals was used to record visitor number (number of adults and children) at each of the four visitor viewing area windows. One-zero sampling for 60 s periods, within each 3 min interval, was used to record visitor behaviour (described in Table 2). This was chosen for practical reasons and previous studies that have investigated the dwell time of visitors at various zoo animal exhibits, found that while there were differences between species, the overall average dwell time for visitors was 30 s when animals were inactive and 60 s when animals were active [44,45]. These observations were made from the top of the visitor viewing area (pass Viewing window 4), to reduce the effect of researcher presence on the penguins.

### 2.5. Statistical Analysis

Each study measurement was calculated as a summary value for each study day to obtain 5 replicate values of each of the two treatments. On each day, the proportion of penguins visible in each area was calculated at each sample point (= number of penguins visible in the area ÷ the number of penguins in the exhibit on that day). These values were averaged over sample points to provide a daily value for the proportion of penguins visible in each area. On each day, the proportion of visible penguins that were displaying a behavioural state in an area was calculated for each sample point (= number of visible penguins in area displaying behavioural state ÷ the total number of visible penguins) and then averaged over the day. Animal behavioural events were handled in a similar manner whereby the proportion of visible penguins displaying each behavioural event in an area within a 30 s period was recorded, and then averaged over each 30 s period in a day. In relation to visitor variables, for each day, the total number of visitors (all ages) observed at the enclosure was calculated by summing the number of visitors at each viewing area window during the six observation blocks. The proportion of the observed visitors at each visitor viewing area window was calculated from these totals. Average ambient noise level (dB) was calculated from the daily trace. The daily visitor behaviour proportions were calculated by summing the number of 60 s periods that each visitor behaviour occurred for each day divided by the total number of 60 s periods per day (i.e., 180) to obtain a single summary value.

Prior to statistical analysis, on each day, the summary values for animal behaviour (state and events), the proportion of visitors at each viewing window and visitor behavioural events were angularly transformed, and the daily values for the number of visitors at the enclosure were square root transformed. This ensured that the residual variation was similar in all treatments, and the distribution of residuals had minimal skewness. No transformation was required for ambient noise level.

All measurements were analysed as a two-treatment analysis of variance (ANOVA) for a fully randomised design (equivalent to an unpaired two-sided t-test), using the daily summary values as the unit of analysis. With the exception of ‘ambient noise level’, this resulted in an analysis with eight residual degrees of freedom. The analysis of variance for ‘ambient noise level’ had seven residual degrees of freedom because on one day the ambient noise level data was unavailable due to a technical issue. Non-parametric permutation tests, based on the treatment effect F-values, were used for some visitor and animal behaviour measures due to the large number of zero values for some behaviours, which can cause over-sensitivity for detecting treatment effects in the parametric analyses. Analyses were carried out using the ANOVA directive and APERMTEST procedure of the GenStat 18 statistical package [46].

## 3. Results

### 3.1. Visitor Variables

When the main visitor viewing area window was covered, the proportion of visitors present at this window decreased by about 85% (F_1,8_ = 117.47, *p* < 0.001) which was mostly balanced by the increase in the proportion of visitors present at Viewing window 2 (F_1,8_ = 43.82, *p* < 0.001; Table 3). 

As expected, covering the main visitor viewing area window eliminated visitors from interacting with penguins, being on the enclosure ledge and passively standing at this window (Table 4). Also, covering the main window decreased the proportion of 60 s sample periods in which visitors made tactile contact with the window, loud vocalisations and sudden movement at this window (*p* < 0.05; Table 4). However, there was no corresponding effect (*p* > 0.05) on ambient noise level in the visitor viewing area, the number of visitors at the enclosure, the proportion of visitors at Viewing window 3 and Viewing window 4 (Table 3) or any visitor behaviours at Viewing windows 2, 3 and 4.

### 3.2. Little Penguin Behaviour

The average proportion of penguins visible in the Corner area, depending on the day, varied from 30 to 60% and the proportions of penguins visible in the area near the main window varied from 10 to 20% (Figure 6). Also, penguins only spent a small proportion of time in Viewing window areas 2, 3 and 4 (Figure 6). Covering the main visitor viewing area window was found to have no effect on the overall proportion of penguins visible except for the proportion of penguins visibly present in the area near the main visitor viewing area window. When the main window was covered, the proportion of penguins present in the area near this main window increased by about 20% (F_1,8_ = 6.05, *p* < 0.05; Table 5). Furthermore, covering the main window reduced the proportion of visible penguins that were vigilant in the area near this window by about 70% (F_1,8_ = 6.18, *p* < 0.05) and increased the proportion of visible penguins preening in the water in the area near this window by about 180% (F_1,8_ = 5.98, *p* < 0.05). An approximate 20% reduction in the proportion of visible penguins’ surface swimming in the Corner area (F_1,8_ = 6.21, *p* < 0.05) was also found when the main visitor viewing area window was covered (Table 6). There were no treatment effects (*p* > 0.05) on the other penguin behavioural states or events or in the areas near the other visitor viewing area windows (Table 6). 

## 4. Discussion

Our present experiment aimed to determine the effect of covering the main visitor viewing area window on little penguin behaviours indicative of fear, such as huddling, avoidance and vigilance. Covering the main visitor viewing area window increased the proportion of penguins present and the visible penguins preening in the water and decreased the proportion of visible penguins vigilant in the area near the main window. This provides some evidence that the little penguins in this exhibit showed an aversion to the area near the main window when it was uncovered, with the reduced presence of the penguins and increased vigilance reflecting increased avoidance of and alertness towards visitors respectively. This suggests unobstructed visual contact with visitors may be fear-provoking for these little penguins. The reduced preening in the water also suggests that contact with visitors may be threatening for little penguins since preening behaviour is a comfort behaviour that is more likely to occur when animals are not threatened [47,48,49,50]. The results in the present study are supported by two experiments on zoo-housed little penguins where the presence and close viewing proximity of visitors increased vigilance and avoidance behaviours and reduced preening in the water in little penguins at Melbourne Zoo [3,10]. Also, this is consistent with Klomp and colleagues [37] and Weerheim and colleagues [34] who found wild little penguins increased their avoidance of areas with high human traffic and footpaths when nesting. Although the effects of visual contact were not specifically examined in these studies, it can be surmised that visual contact may be a key component to little penguin fear of visitors. However, it is important to note that vigilance is not always an indicator of fear and may indicate curiosity and since the main window was randomly covered for five 1-day periods over 2 weeks, the novelty of covering the window and the associated changes such as illumination in the area, may have stimulated inspection [1,51]. Consequently, this may explain the increased presence of penguins in the area near the main window during this treatment imposition. Zoo-housed little penguins have been found to show curiosity to novelty, particularly to mirrors likely due to the reflective properties appealing to them [52]. This may be true in the present experiment where covering the main visitor viewing area window using the barrier may have increased the reflective properties of this window. However, this requires further research to tease out the effects of novelty of an exhibit manipulation and its potential effects on penguin responses to visitors.

Covering the main visitor viewing area window decreased all visitor behaviours at this window that may be fear-provoking for penguins such as tactile contact with the glass viewing window, loud vocalisations and sudden movement. This is to some extent supported by previous studies that, while not measuring visitor behaviour specifically, found camouflage netting and one-way vision screens improved gorilla and black-capped capuchin welfare through the reduction of visual contact with visitors but also likely by reducing potentially threatening visitor behaviours as a result of the visual barriers being in place [22,23,24]. Our present experiment also found that when the main window was covered, the overall number of visitors at the enclosure did not change but did reduce the proportion of visitors present at the main window. The number of visitors at the main window was balanced by the increase in the number of visitors present at the adjacent viewing window, Viewing window 2. A possible explanation may be that visitor dwell time and visitor viewing experience were not adversely affected by the visual barrier, but this requires further investigation. This contrasts with previous studies on little penguins that had reductions in visitor numbers when barriers were used to regulate visitor-animal interactions [3,23]. 

Little penguins, like all penguin species, have very good vision as they are visual predators that rely on their vision to find and catch prey underwater [18,53]. Also, being the smallest and only penguin species whose activity on land is strictly nocturnal, having good vision is important for this species [18,20]. Good vision aids in the avoidance of predators especially for little penguins who have terrestrial, aerial and aquatic predators [16,17,18,19]. Consequently, it would not be surprising that visual contact with visitors affected avoidance, vigilance and preening behaviour in little penguins in our present experiment. However, it is unclear in the present experiment whether these behavioural changes in penguins when the main window was covered were due to simply eliminating visual contact with visitors at this window *or* reducing other aspects of visitor contact such as potentially fear-provoking visitor behaviours. In the present experiment, the visual barrier placed on the main visitor viewing area window reduced all potentially fear-provoking visitor behaviours such as tactile contact with the glass window, ground vibrations created by visitors from sudden movement and loud vocalisations. These visitor-induced sounds and vibrations in the water may be fear-provoking for little penguins. Research has found both substrate-borne vibrations and water-borne sounds can affect the behavioural responses of many marine species, particularly fish, and can be an important source of information for predator avoidance or prey detection [54,55,56,57]. Water is also an effective medium for sound propagation as a result of its low absorption rates and low attenuation which allows sound to travel five times faster in water compared to air [57]. Thus, in addition to visual contact with visitors, visitor-induced sounds and vibrations may have been fear-provoking behaviours for the study penguins resulting in both increased avoidance and vigilance and reduced preening in the water in the area near the main window when the visual barrier was not in place. Interestingly, Higham and Shelton [58] found wild little penguins have an intrinsic tolerance to boat engine vibration and noise where the authors suggested that this shows little penguins can habituate to human presence. However, vibrations and noise cause by mechanical equipment may be perceived differently by little penguins compared to vibrations and noise caused by humans directly. Further research is required to understand the effects of visitor-induced sounds and vibrations on little penguin fear responses as studies have shown that regulating close visitor contact and potentially fear-provoking visitor behaviours using management strategies, such as effective signage or barriers, can reduce the adverse effects of visitors on animals [3,22,23,59,60].

The high proportion of visible penguins in the Corner area whether the main visitor viewing area window was covered or uncovered indicates that the penguins had a preference for the Corner area in comparison to other areas of the exhibit. The Corner area allowed the penguins to be visually hidden from visitors by the solid wall of the pool and therefore this area may have provided the penguins with an opportunity to avoid visitors and reduce the perceived close proximity of visitors which may be something zoos should consider for their animals. This further supports our interpretation that visitors may be fear-provoking for little penguins in this exhibit. Research has shown that an area in which to retreat may minimise visitor-induced stress on zoo animals by providing a level of choice and control over their interactions with visitors [61,62,63,64]. Similarly, increased separation between visitors and penguins has shown to minimise the effects of visitors on little penguin fear responses [3]. Consequently, penguins in zoos may benefit from ameliorations such as one-way viewing glass, barriers to close visitor contact and areas for penguins to retreat. However, this Corner area may have been also attractive to the penguins because it was the deepest part of the pool and was located next to the breeding group of adult penguins, where the two groups had visual contact with each other. Little penguins are a highly social species and in the wild live in colonies like all penguin species and thus, it is not surprising that social contact with the adult penguins may have attracted the penguins to the Corner area [65].

The apparent fewer visible penguins surface swimming in the Corner area is anomalous because it contradicts previous findings that close contact with visitors reduces surface swimming. Sherwen and colleagues [10] found that closing the exhibit to visitors increased surface swimming and Chiew and colleagues [3] found that increased separation between visitors and penguins increased surface swimming in little penguins. In the present experiment when penguins were in the Corner area, they could not see visitors at the main visitor viewing area window so this would have expected to result in an increase in surface swimming from the lack of visual contact with visitors. However, it is not clear why surface swimming was affected by treatment in the Corner area in the opposite direction to what has been previously found and since it was unaffected in the area near the main window. 

The findings from the present experiment at Taronga Zoo support those from the two previous experiments at Melbourne Zoo by the authors that some types of visitor contact may be fear-provoking [3,10]. While the present experiment found some relatively moderate effects of reduced visitor contact on avoidance and vigilance behaviour, the two experiments at Melbourne Zoo found strong effects of reduced visitor contact on avoidance, huddling and vigilance behaviour, as well as surface swimming. It is not surprising that there are some differences in the magnitude of the treatment effects between these experiments since the treatments differed substantially. In particular, the experiments at Melbourne zoo examined the effects of closing the enclosure to visitors and regulating visitor viewing proximity on the penguins, while the present experiment examined the effect of covering the main visitor viewing area window, but not the other three visitor viewing area windows. Furthermore, there were differences between experiments in keeper management practices, enclosure design and thus the type of possible visitor interactions such as close contact and visitor encounters, as well as individual animal characteristics such as past experiences and coping style. For example at Melbourne Zoo, visitors were able to view the penguins from most of the perimeter of the enclosure and along the main length of the pool where there were opportunities, if visitors chose, to make tactile contact with the penguin pool or the penguins themselves by looming over the pool edge. In the present experiment, the enclosure is long and thin with a long water course and elongated stretch of land and the visitor viewing area was limited to one side of the enclosure where visitors were primarily positioned below the penguins, viewing the penguins mostly under water through the glass windows. This enclosure design may have helped ameliorate any fear-provoking aspects of visitors. The little penguins in the present study also may have had more opportunity to habituate to visitors because unlike the penguins at Melbourne Zoo, these penguins had daily visitor encounters [41,42,66,67]. However, considering the behavioural changes in the present study, it is possible that the behavioural response towards visitors by penguins during a closeup encounter, involving controlled visitor-penguin interactions, may differ from their responses to visitors normally viewing them on exhibit from the visitor viewing area where visitors can be unpredictable and intense and thus, are uncontrolled visitor-penguin interactions. 

An obvious question arising from these three experiments is the implications of the customary visitor contact on the welfare of penguins at these two enclosures. Chiew and colleagues [3] found that either the installation of a physical barrier or by closing the exhibit to visitors strongly reduced little penguin behaviours indicative of fear, but did not affect faecal glucocorticoid metabolites. The authors suggested that although fear responses to visitors were observed when visitor viewing proximity and behaviour were uncontrolled which is a welfare concern, these fear responses such as avoidance may have been an effective adaptive response in ameliorating the activation of the hypothalamo–pituitary–adrenal (HPA) axis. The authors also suggested that the treatment period may have been insufficient to elicit a measurable change in faecal glucocorticoid metabolites concentrations (i.e., sustained activation of the HPA axis). Therefore, avoidance of the area near the main window by the penguins in the present study, as well as preference for the Corner area of the pool, which was hidden from visitors, may have been successful in reducing the penguins’ welfare risks from visitor contact at the main window. Clearly, further research on the effects of the nature and intensity of visitor contact on both the behaviour and stress physiology of little penguins would be prudent. For example, this could entail systematically varying enclosure design or imposing treatments for more than one or two-day periods and measuring the effects on animal behaviour and other physiological measures such as faecal glucocorticoid metabolites and heart rate. 

## 5. Conclusions

This experiment provides limited evidence that little penguins in this exhibit showed an aversion of the area near the main visitor viewing area window when it was uncovered, suggesting that visitors may be fear-provoking for these little penguins. However, it is unclear whether the apparent aversion of this window when uncovered was due to simply having visual contact with visitors and/or other aspects of visitor contact occurring at the window such as potentially fear-provoking visitor behaviours. This interpretation is supported by the observation that the penguins spent most of their time in the Corner area where visitors were not visible to the penguins. Therefore, the welfare implications of leaving the main window uncovered on these little penguins may have been minimal but it does suggest that penguins in zoos may benefit from modifications to the enclosure that may ameliorate penguin fear responses to visitors such as one-way viewing glass, barriers to close visitor contact and areas for penguins to retreat.

## Figures and Tables

**Figure 1 animals-10-01224-f001:**
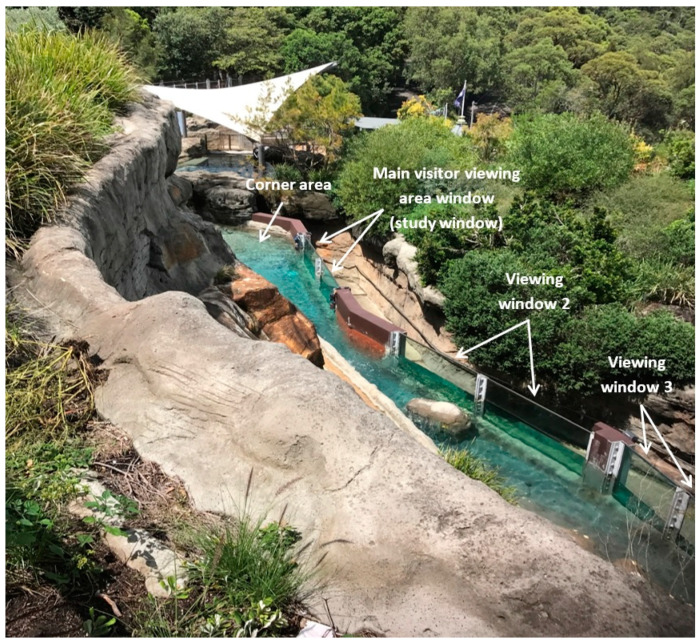
The Great Southern Oceans penguin exhibit at Taronga Zoo. Focus area was the ‘Encounter section’ of the exhibit (long, skinny pool section in the above figure) where the little penguins involved in closeup encounters were located. The main breeding group of little penguins were located adjacent to this section (i.e., underneath the white sail). The two groups had no access to each other.

**Figure 2 animals-10-01224-f002:**
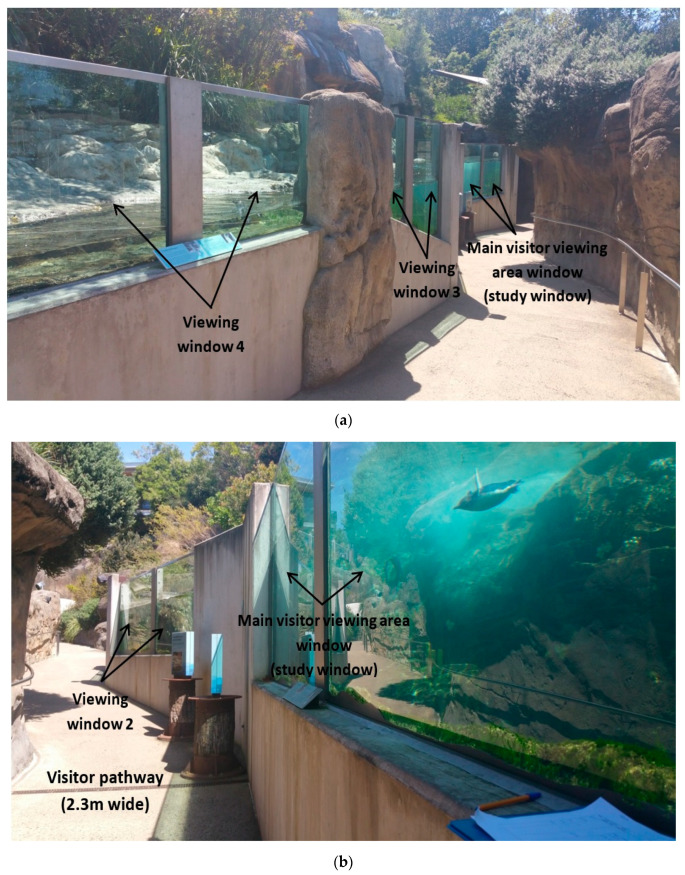
(**a**)**.** The outdoor encounter section of the penguin exhibit that consisted of four two-pane glass visitor viewing area windows (respectively labelled Main visitor viewing area window and Viewing window 2, 3 and 4) and a visitor pathway that slopes down to an underground viewing area of the main breeding group of penguins. (**b**)**.** The main visitor viewing area window and Viewing window 2 of the encounter section of the penguin exhibit where the main visitor viewing area window was our study window.

**Figure 3 animals-10-01224-f003:**
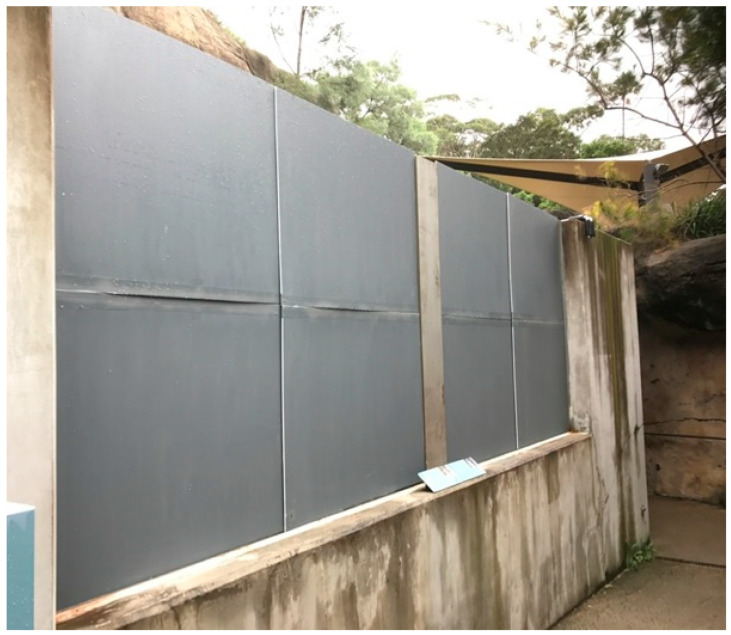
The visual barrier used to block contact between visitors and penguins at the main visitor viewing area window.

**Figure 4 animals-10-01224-f004:**
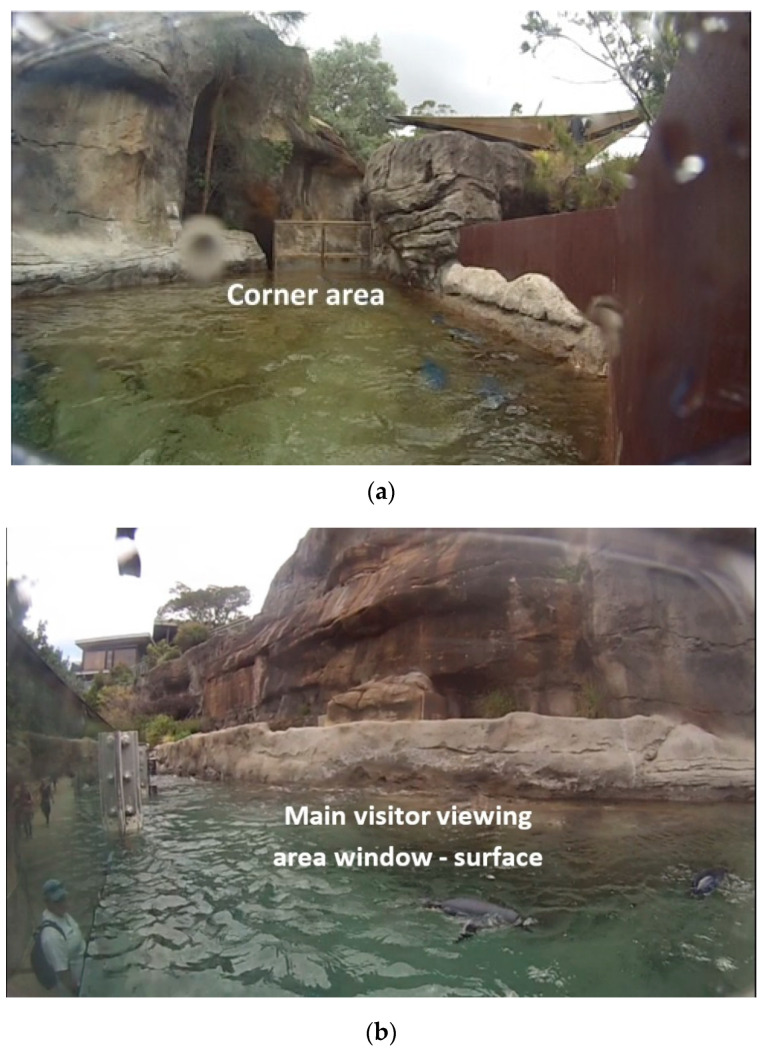
Camera fields of view (**a**). ‘Corner’ area (not visible to visitors); (**b**). ‘Main visitor viewing area window’ area (above water); (**c**). ‘Main visitor viewing area window’ area (underwater).

**Figure 5 animals-10-01224-f005:**
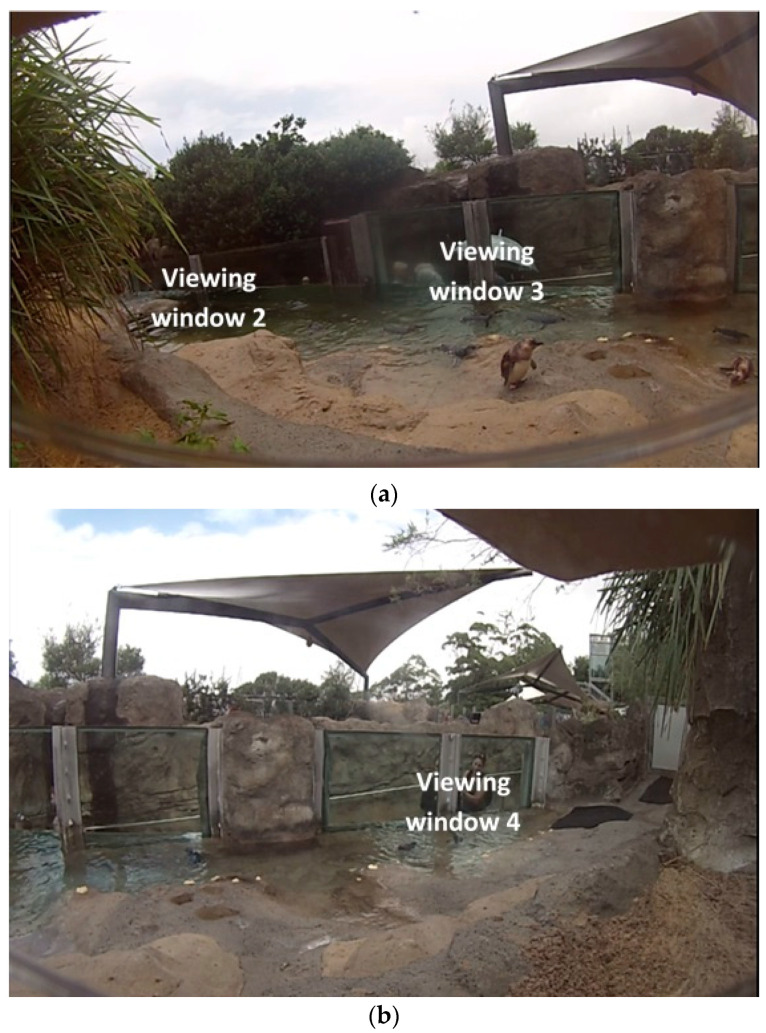
Camera fields of view (**a**). Viewing window 2 and 3 areas; (**b**). ‘Viewing window 4’ area (where keepers entered and exited).

**Figure 6 animals-10-01224-f006:**
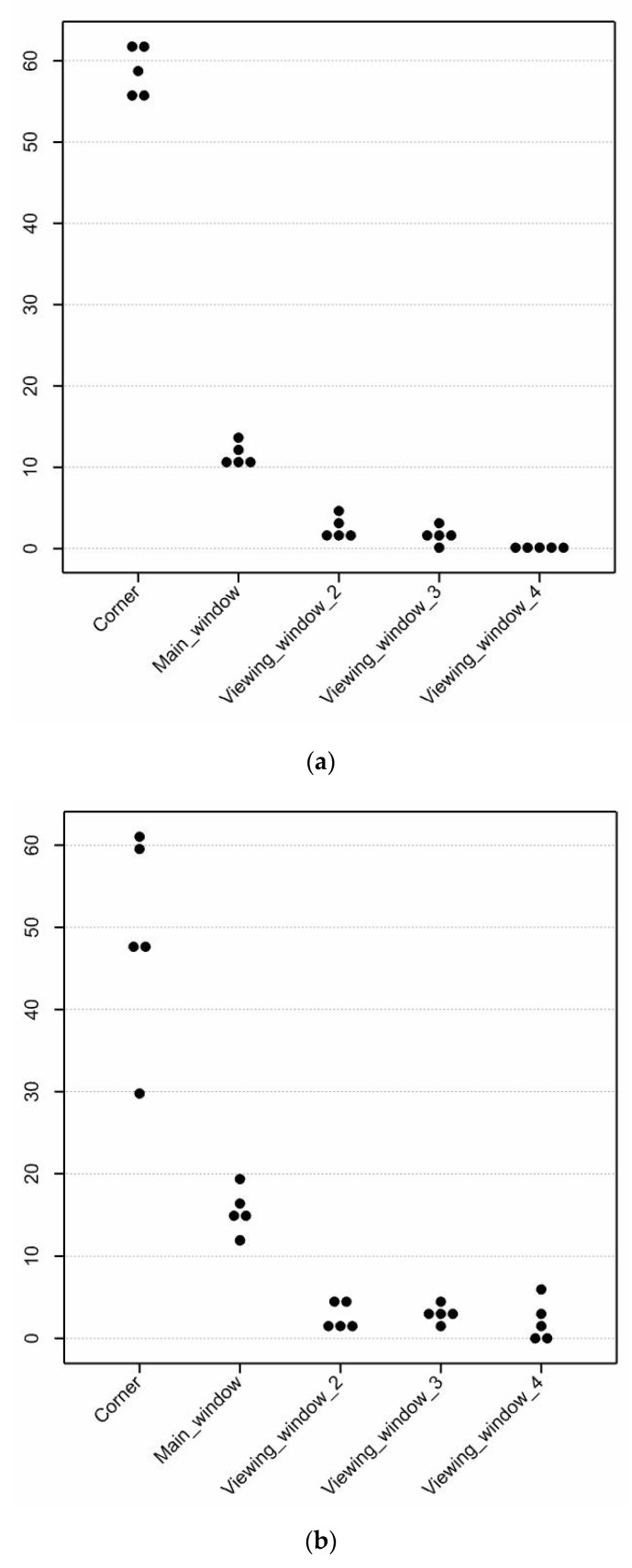
Dot histograms of the average proportion of penguins visible (%) in each area near the four visitor viewing area windows and the corner with the (**a**) main window uncovered and (**b**) main window covered. Each dot represents observations from a single day.

**Table 1 animals-10-01224-t001:** Ethogram of penguin behaviour. Adapted from Chiew et al. [3].

Behaviour	Description
*States*	
Penguins visible	Penguins present in the field of view of the camera(s).
Huddling (land)	Stationary, positioned within one flipper distance of at least one other penguin.
Resting (land)	Belly on the ground, in a prone position, with eyes open or closed.
Idle (land)	Standing on two feet with a relaxed posture, eyes open or closed; not visually scanning the environment.
Locomotion (land)	Upright position, moving from one location to another either by walking or running.
Vigilance (land)	Standing on two feet, visually scanning the environment with head movements from left to right or vice versa.
Surface swimming	Moving or floating on the surface of the water, with head erect or in the water.
Diving	Locomoting under water surface.
*Events*	
Preening (on land)	Standing upright on two feet, running bill through plumage.
Preening (in water)	On the surface of the water and running bill through plumage.
Allopreening	Running bill through the plumage of another bird(s).
Porpoise	Continuous swimming in rhythmic serial leaps where the whole-body leaps in and out of the water’s surface repeatedly.
Fast swimming	Sudden, rapid movement under water.
Agonistic interactions (one or a combination of these behaviours)	Peck: Directed at another individual in which an individual directly hits or strikes at another bird with its bill. Bill slap: hit or strike with the side of the bill. Bill joust: an individual interlocks bill with another individual’s bill. Lunge: sudden forward thrust of the body towards another individual. Chase: an individual runs or swims after another individual.
Interaction with glass	Head orientated towards the glass window, either pecking with bill and/or swimming directly at the glass window.
Manipulate object	Using bill to peck or nibble an inanimate object(s) such as rocks, vegetation (e.g., straw or grass) or camera case.
Bathing	In the water, twisting body from side to side in a shaking motion.
Other	Any other behaviour not described above e.g., flapping one or both wings, chasing insect.

**Table 2 animals-10-01224-t002:** Ethogram of visitor behaviours adapted from Chiew et al. [3].

Behaviour	Description
Tactile contact with glass windows	Tapping or banging on the glass viewing windows using hands or fingers.
Loud vocalisations	Shouts, screams, loud whistles to attract animals’ attention.
Sudden movement	Running, waving or jumping towards or at the penguin(s) and/or exhibit.
Interaction with a penguin(s)	Interacting with a penguin(s) whereby the penguin(s) is following visitor hand movements or presence.
On enclosure ledge	Standing, sitting and/or climbing on enclosure ledge. May or may not be leaning against the glass visitor viewing window(s).
Passively observing	Standing back, quiet/silent while looking at penguin enclosure. No sudden movements.
Other	Any behaviour not described above e.g., copying penguin vocalisations and flash photography.

**Table 3 animals-10-01224-t003:** Effects of treatment on ambient noise level and visitor numbers at the penguin enclosure. The means are the mean noise level and number of visitors at the enclosure each day (per six 30 min observation blocks each day). The proportion of visitors at each visitor viewing area window are also shown. Back transformed means are presented in parentheses and any statistical significance (*p* < 0.05) are bolded.

Visitor Variables	Main Window Uncovered	Main Window Covered	s.e.d.^c^	*p*-Value ^a^
Ambient noise level (dB)	59	60	0.70	0.38 ^b^
Visitors at the enclosure *	15 (228)	13 (181)	1.5	0.28
% Visitors at the main visitor viewing area window **	40 (40)	15 (6.3)	2.3	**<0.001**
% Visitors at Viewing window 2 **	30 (25)	46 (52)	2.5	**<0.001**
% Visitors at Viewing window 3 **	21 (13)	22 (14)	2.8	0.66
% Visitors at Viewing window 4 **	27 (21)	31 (27)	2.8	0.23

* Visitors at the enclosure was square root transformed. ** The proportion of visitors at each glass viewing window were angularly transformed ^a^
*p* values were calculated using F tests based on one, eight degrees of freedom except for ‘Ambient noise level’ where the *p* value was calculated using F tests based on one, seven degrees of freedom (due to 1-day of missing data). ^b^
*p* values calculated using permutation test. ^c^ s.e.d denotes standard error of difference.

**Table 4 animals-10-01224-t004:** Effects of treatment on the mean proportion of 60 s sample periods in which visitor behaviours occurred at the main visitor viewing area window (observed in six 30 min observations blocks: total of 180 sample periods each day). Data were angularly transformed. Any statistical significance (*p* < 0.05) are bolded.

Visitor Behaviour	Angularly Transformed			Back Transformed (%)		*p*-Value ^a^
	Main Window Uncovered	Main Window Covered	s.e.d ^b^	Main Window Uncovered	Main Window Covered	
Tactile contact with glass windows	19	0	2.1	11	0	**<0.001**
Loud vocalisations	17	8.1	2.0	8.9	2.0	**<0.01**
Sudden movement	14	5.7	2.6	5.4	0.99	**<0.05**
Interaction with a penguin(s)	13	0	2.3	4.8	0	**<0.001**
On enclosure ledge	11	0	3.4	3.6	0	**<0.05**
Passively observing	7.7	0	2.8	1.8	0	**<0.05**

^a^*p* values were calculated using F tests based on one, eight degrees of freedom. ^b^ s.e.d denotes standard error of difference.

**Table 5 animals-10-01224-t005:** The effect of treatment on the proportion of penguins visible (%) overall and the proportion of penguins visible in each area near the four visitor viewing area windows and the ‘Corner’. Data were angularly transformed. Any statistical significance (*p* < 0.05) are bolded.

	Angularly Transformed			Back Transformed (%)		*p*-Value ^a^
	Main Window Uncovered	Main Window Covered	s.e.d ^b^	Main Window Uncovered	Main Window Covered	
Penguins visible overall	60	58	2.5	74	72	0.60
Corner	50	45	3.4	59	49	0.14
Main visitor viewing area window	20	23	1.2	12	15	**<0.05**
Viewing window 2	8.5	9.2	1.4	2.2	2.6	0.60
Viewing window 3	6.7	9.6	1.4	1.4	2.8	0.077
Viewing window 4	2.9	6.5	2.5	0.26	1.3	0.19

^a^*p* values were calculated using F tests based on one, eight degrees of freedom. ^b^ s.e.d denotes standard error of difference.

**Table 6 animals-10-01224-t006:** The effect of treatment on the proportion of visible penguins in each behaviour (states and events) in the area near the main visitor viewing area window and the ‘Corner’. Data were angularly transformed. Any statistical significance (*p* < 0.05) are bolded.

Behaviour	Angularly Transformed			Back Transformed (%)		*p*-Value ^a^
	Main Window Uncovered	Main Window Covered	s.e.d ^c^	Main Window Uncovered	Main Window Covered	
**Corner (not visible to visitors)**						
*States*						
Surface swimming	59	50	3.9	74	58	**<0.05**
Diving	10	14	1.8	3.3	6.1	0.068
*Events*						
Preening (water)	26	25	2.2	19	17	0.66
Allopreen	1.4	1.4	0.37	0.062	0.060	0.98 ^b^
Agonistic interactions	3.4	2.9	0.79	0.36	0.25	0.53 ^b^
Porpoise	4.1	4.0	0.67	0.51	0.49	0.88 ^b^
Fast swimming	7.5	7.2	0.73	1.7	1.6	0.77 ^b^
Interaction with glass	0	0	-	0	0	1.0
Manipulate object	0.40	0.95	0.51	0.0049	0.027	0.41 ^b^
Bathe	3.1	4.9	2.4	0.29	0.73	0.48
Other	2.4	2.3	0.68	0.18	0.16	0.88 ^b^
**Main visitor viewing area window**						
*States*						
Huddling	6.0	3.6	2.1	1.1	0.39	0.30
Resting	0	0	-	0	0	1.0
Idle	5.9	5.5	1.5	1.1	0.93	0.81
Locomotion	0.41	0.40	0.57	0.0051	0.0049	1.0 ^b^
Vigilant	5.2	3.0	0.88	0.82	0.28	**<0.05** ^b^
Surface swimming	15	18	2.0	6.7	9.2	0.24
Diving	16	20	2.8	7.9	11	0.29
*Events*						
Preening (land)	6.6	6.6	1.3	1.3	1.3	0.99
Preening (water)	3.3	5.5	0.90	0.33	0.92	**<0.05** ^b^
Allopreen	0.63	1.2	0.51	0.012	0.040	0.40 ^b^
Agonistic interactions	2.1	2.3	0.59	0.13	0.16	0.75 ^b^
Porpoise	3.5	4.3	0.98	0.37	0.56	0.44 ^b^
Fast swimming	5.4	6.5	1.1	0.90	1.3	0.33
Interaction with glass	9.1	13	2.2	2.5	5.1	0.12
Manipulate object	6.5	7.3	1.6	1.3	1.6	0.61
Bathe	0.75	1.6	0.61	0.017	0.075	0.24 ^b^
Other	0.62	0.27	0.38	0.012	0.0022	0.52 ^b^

^a^*p* values were calculated using F tests based on one, eight degrees of freedom. ^b^
*p* values calculated using permutation test. ^c^ s.e.d. denotes standard error of difference.

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
