# Peer review of "Effect of Covering a Visitor Viewing Area Window on the Behaviour of Zoo-Housed Little Penguins (*Eudyptula minor*)"

_animals, 2020, doi:10.3390/ani10071224_

Round 1

Reviewer 1 Report

animals-penguins-862149

In this manuscript, the authors investigated the effect of covering a visitor viewing area window on fear behaviour of zoo-housed little penguins. They found that the ‘visitor effect’ is negative for this species, thus visitors may be fear-provoking for zoo-housed little penguins. In general, the study is clear and well-motivated, even if there are very few days of the experiment (and also the number of penguins observed is small) and this can lead to problems from a statistical point of view. However, this is an interesting study, which can be preliminary to further studies with a greater number of animals and in different enclosures.

Please see below for a few comments and suggestions that the authors could consider for revisions:

Lines 21-23 I suggest removing this sentence

Line 37 I don’t agree, it could be an animal welfare problem, see the Five Freedom of animal welfare

Overall, The authors should use a consistent style for p-values. Hereafter, I suggest reporting p-values using the APA style, e.g., p<0.001 instead p= 0.000005

Line 85 I suggest adding this reference on visitor effect on penguins https://pubmed.ncbi.nlm.nih.gov/26673870/

Line 89 I suggest Chiew et al.

Lines 149-151 This is not clear to me, are there only 13 penguins that have access to the windows?

Line 151-154 how long the experiment lasted? Please indicate the start date and the end date. I suggest adding this information and the total number of days of the experiment (including the total number of treatment) here.

Lines 156-159 I suggest adding the total area of the enclosure, and the area of the swimming pool

Caption figure 2:  I suggest removing ‘Picture of’

Line 185: I suggest ‘Sherwen et al.’

Line 228 I suggest adding this reference for the sampling methods (Altman, 1974) https://www.jstor.org/stable/4533591?seq=1

Caption Table 2 and 3: Chiew et al

Line 282 Authors have chosen to use ANOVA to compare the daily summary values, even if they have few days of observation. Could you please justify this choice? Because if your data are clearly non-normal then you should consider using the Friedman test, a nonparametric alternative.

Have the authors observed differences between males and females?

Lines 365 -366 I suggest Klomp et al. and Weerheim et al.

Reviewer 2 Report

I have made some minor suggestions, with references, in the attached file.

Reviewer 3 Report

Overall, this is a well executed study baring the standard limitations that are inherent in zoo research, namely study duration and sample size. Although not significantly novel compared to the original study, with some minor adjustments, this manuscript should be considered for inclusion in the special issue.

My primary concerns are with the fact that the study group had “daily interactions with visitors”, which presumably could be explored further as a factor on the behavioral responses, at the very least temporally, during the study period. A deeper explanation of how and when those interactions occurred (if they did during the study period), and how those affected the observations at the time points would be helpful. Assuming the third and fourth observation time blocks are unique to accommodate the guest feeding (again if it was offered during the study), a pre and post behavioral analysis could help clear up the response. The Discussion addresses the point that the spatial differences are significant enough to have little impact on behavior, but attention to the temporal component would strengthen the paper.

It would be prudent to identify or explain the particular breeding phase of the adjacent penguin group as breeding vocalizations could be drawing particular individuals to “the corner” that is closest in proximity. Again this is mentioned in the discussion, but a more comprehensive look at the potential attractions could aid in further experiments. 

An explanation of the location of the sound recording devise would make the findings of the sound study more robust, assuming it was collecting data from inside the exhibit from the penguins “perspective”. Regardless, it should be described better.

The novelty of the barrier is also a concern as some habituation could have occurred by the fifth deployment. I recognize that the numbers are small, but was there any longitudinal trends worth noting that make this information valuable?

Figure 6. seems redundant and perhaps not needed in its current form.

Finally, the discussion reads more like a review. Although it is appreciably critical and well thought out, perhaps the language could be shifted to a more applied tone, with replication or advanced techniques in mind.  
